# Comparison of Platelet-Rich Plasma Treatment and Partial Plantar Fasciotomy Surgery in Patients with Chronic Plantar Fasciitis: A Randomized, Prospective Study

**DOI:** 10.3390/jcm11236986

**Published:** 2022-11-26

**Authors:** Ran Atzmon, Dynai Eilig, Jeremy Dubin, Matias Vidra, Omer Marom, Alex Tavdi, Michael Drexler, Esequiel Palmanovich

**Affiliations:** 1Assuta Medical Center, Department of Orthopaedic Surgery, Affiliated with the Faculty of Health and Science and Ben Gurion University, Ha-Refu’a St. 7, Ashdod 7747629, Israel; ranatzmon@gmail.com (R.A.); ranatzmon1@gmail.com (D.E.); michaeldrexler@gmail.com (M.D.); ezepalma@gmail.com (E.P.); 2Tel Aviv Medical Center, Department of Orthopaedic Surgery, Affiliated with the Sackler Faculty of Medicine and Tel Aviv University, Weizmann St. 6, Tel Aviv 6423906, Israel; matiasvidra@gmail.com; 3Orthopaedic Department, Meir Medical Center, Kfar Saba, Affiliated to Sackler Faculty of Medicine, Tel-Aviv University, Tel Aviv 6423906, Israel; dubinjeremy1@gmail.com (O.M.); alextavadii@gmail.com (A.T.)

**Keywords:** platelet-rich injection, chronic plantar fasciitis, steroid injection, partial plantar fasciotomy

## Abstract

Platelet-Rich Plasma (PRP) injection has become a desirable alternative to Partial Plantar Fasciotomy (PPF) surgery and steroid injection for patients with chronic plantar fasciitis (CPF) due to its potential for shorter recovery times, reduced complications, and similar activity scores. As such, we compared PRP treatment to PPF surgery in patients with CPF. Between January 2015 and January 2017, patients were randomly divided into two groups, a PRP treatment group, and a PPF group. All procedures were performed by a single foot and ankle fellowship-trained specialist surgeon. Visual Analog Score (VAS) and Roles-Maudsley Scale (RM) were collected during the preoperative visit and 3, 6, and 12 months postoperatively. The patients were also closely followed by a physiotherapist. There were 16 patients in each group after four patients refused to participate. Patients in the PPF had low Roles-Maudsley Scale (RM) scores compared to the PRP group one-year after treatment (3.77 vs. 2.72, *p* < 0.0001). Both procedures showed a reduction in RM scores during the follow-up year (9 to 1.62 for PPF and 8.7 to 2.4 for PRP). There was no significant change in VAS pain between the two groups (*p* = 0.366). Patients treated with PRP injection reported a significant increase in their activity scores, shorter recovery time, and lower complication rates compared to PPF treatment. Moreover, with respect to existing literature, PRP may be as efficient as steroid injection with lower complication rates, including response to physical therapy. Therefore, PRP treatment may be a viable option before surgery as an earlier line treatment for CPF. Level of Clinical Evidence: II.

## 1. Introduction

Chronic Plantar Heel Pain (CHPN) is one of the most common foot disorders encountered by physicians in their clinics, accounting for approximately 15% of all foot complaints in the adult population., while Plantar Fasciitis (PF) has been amongst the leading causes of CHPN. Although Plantar Fasciitis is multifactorial in nature, microruptures have been described as a dominant contributing factor to pain due to collagen necrosis and fibro-fatty tissue formation [1]. In addition to microruptures, other risk factors for PF include obesity, decreased ankle dorsiflexion, heel spurs, anatomical variations such as cavus foot, inadequate supporting shoes, and occupational limitations that require prolonged standing [1,2].

Plantar fasciitis can be divided into three groups according to the onset of pain: Acute PF (4–6 weeks), Sub Acute PF (6–12 weeks) and Chronic PF (>3 months). Chronic Plantar Fasciitis (CPF) can be further divided into the refectory and recalcitrant periods, with the latter consisting of symptoms presiding more than six months without improvement after appropriate treatment [3]. PF is most commonly diagnosed clinically or using Ultra Sound (US) imaging that may show hypoechoic fascia and increased thickness (>4.5 mm) [4]. Ninety percent of the patients will respond well to conservative treatment within nine months from the first onset of pain [5]. Some of the most readily used conservative treatments described in the literature include; physical therapy, orthotic devices, splitting and walking casts, oral Nonsteroidal Anti-Inflammatory Drugs (NSAID), corticoid steroids injection, shock wave therapy and Platelet-Rich Plasma (PRP) injection. Usually, surgical intervention such as Partial Plantar Fasciotomy (PPF) is considered when conservative treatment fails. Although surgical intervention has yielded good postoperative outcomes (83–89% success rate), surgery can also come with certain postoperative risks [6].

Platelets-Rich Plasma (PRP) injection, also known as pure PRP (P-PRP) injection, is an emerging therapy to treat persistent joint inflammation through anti-inflammatory vascularization as well as angiogenesis derived from platelets [7,8,9,10,11,12]. In this randomized, prospective study, we sought to compare PRP injection versus PPF for patients with recalcitrant CPF in terms of patient-reported functional and pain outcome measurements. In addition, we hypothesized that PRP treatment would have comparable functional outcomes and pain levels compared to PPF for recurrent CPF.

## 2. Methods

After approval from the local institutional review board, we performed a prospective cohort study from a single institution. From January 2015 to January 2017, patients were randomly divided into two groups consisting of 16 patients per group for each treatment modality. Patients in group one were treated with a PRP injection while patients in group two were treated with a partial plantar fasciotomy (PPF) surgery, with all procedures performed by a single foot and ankle fellowship-trained specialist surgeon. All the patients were briefed about the study’s randomization process and consented to the different procedures. Only after the patient was admitted to the hospital and before taking him to the operating room did the surgeon open a sealed envelope given by the research coordinator allocating the patient to one of the two procedures. A posthoc power analysis was performed to determine the appropriate sample size based on the change in Roles-Maudsley Scale (RM) from pretreatment to post-treatment time points. The RM scale is a subjective four-point assessment of pain and limitations of activity with 1 = excellent result with no symptoms following treatment and 4 = poor, symptoms identical or worse than pretreatment [13]. Inclusion criteria mandated that all patients would be diagnosed clinically for recurrent CPF following conservative treatment, which includes stretching, chiropractic therapy, strengthening, orthotics, and acupuncture for at least 3 months prior to the PRP treatment. Patients were excluded if they had been previously treated with local PRP injection for CPF, underwent previous ankle or foot surgery, received local steroids injection within the last six months, or had other foot pathologies such as fractures, arthritis, or bursitis (Figure 1). Additional exclusion criteria included patients diagnosed with CPF but failed to achieve a minimum of 3 months of conservative treatment and patients who encountered trauma after the treatment or were suspected of benefiting from secondary gain (i.e., active lawsuit). (Figure 2). The patient selection process is included (Figure 3).

In addition, the surgeon assessed each patient using the Visual Analog Score (VAS) before and after the treatment and assessed at 3,6 and 12 months after interventions at the outpatient clinic. The visual analog scale (VAS) is a continuous subjective measurement for acute and chronic pain comprised of a horizontal or vertical line ranging from no pain to extreme/worse pain, which can also be quantified by ascending numbers from 0 to 10 or 100 [14].

These patient-reported outcome scores were used due to their well-established reputation and widespread use [13,14]. Before being treated, the patients were examined and evaluated, looking at their general appearance, heel tenderness, pain, foot shape and arching, and other possible pathologies. Chronic plantar fasciitis was typically diagnosed based on clinical findings and the use of ultrasound (per our long-standing hospital protocol). A diagnosis was made if the ultrasound demonstrated increased thickness above 4 mm of the plantar fascia after excluding other foot pathologies such as arthritis, bursitis, and fractures. Though the Gastrocnemius contracture and range of motion were also assessed, it was decided not to include it in the study due to previous treatments and a lack of comparative baseline information. Hence, we felt this type of information might have a detrimental effect on the results.

## 3. PRP Injection Technique

The PRP was prepared and produced in the outpatient clinic and injected in the same setting. First, 60 mL of venous blood was taken from each patient and mixed into a tube containing 11 mL of Estar Tropocell ^®^. Next, the tube was centrifuged in 250 g, 15 RPM/RCF (*100) for 10 min, using the Rotofix 32A Hettich centrifuge without a supplementary activation agent. As a result, the blood was divided into its basic components (low cell plasma, platelet, red blood cell), and 4 cc of the upper layer containing the Plasma. The platelets were harvested using a sterile vacuumed polypropylene cone tubes. In order to avoid activation and dilution of the PRP, there was no use of anesthesia prior to the tissue encounter. At that point, a single ultrasound-guided PRP injection was injected into the insertion of the plantar fascia at the anterior-middle aspect of the heel [15].

## 4. Partial Medial Fasciotomy Surgical Technique

Local skin sterilization and light sedation preceded the surgery followed by an ankle block. A partial medial fasciotomy was performed by a foot and ankle specialist. The plantar fascia was palpated medially and distally to the calcaneal spur, followed by an oblique incision of 3 cm and a blunt dissection to separate the plantar fascia from the surrounding subcutaneous fat. The fascia was then fully dissected through a small transverse stab incision 3 cm in length just distal to the calcaneal fat pad, which minimizes scarring since it is in line with the skin tension lines. The digits were dorsiflexed, and one-third of the medial plantar fascia was released. At the end of the procedure, the medial fascia was plucked to verify adequate release. The skin was then closed using non-absorbable sutures.

## 5. Post-Treatment Rehabilitation and Evaluation

All patients in both groups received the same post-treatment protocol regardless of the procedure, except for heel-raising insoles that were not allowed in the plantar fasciotomy group. The post-treatment protocol included: ice compression therapy for the first 24 h, immediate weight-bearing and eccentric calf starching exercises as tolerated, and heel-raising insoles in the PRP group that was dictated by the surgeon and physiotherapist [5]. In addition, physical therapy was instituted immediately after the procedure for a minimum of two weekly sessions for 6 weeks. The immediate postoperative treatment focused on restoring the passive range of motion and Achilles tendon lengthening, followed by active motion with gradual advancement to strengthening exercises.

All patients were evaluated in the outpatient clinic at two weeks, three months, six months, and 1-year postoperatively. The foot and ankle surgeon performed a clinical examination, including the patient outcome measurements and a complete evaluation by the physiotherapist.

## 6. Statistical Analysis

The statistical data was processed using an Excel 2013 file (version 15.0, Windows 10). The statistical analysis was performed using Fisher exact tests for categorical variables, Chi-squared test, and Mann–Whitney non-parametric test at a significance level of *p*-value < 0.05 as one-sided tests. Continuous variables were expressed as a mean and standard deviation, and nominal variables were expressed by numbers and percentages. Statistical analysis was performed using IBM Statistical Package for the Social Sciences (SPSS) Statistics, version 25 for Windows (SPSS, Chicago, IL, USA).

## 7. Results

Between January 2015 and January 2017, forty-two patients with CPF were treated, out of which 36 met the inclusion criteria. As previously explained, the patients were randomly divided into two equal-sized groups. This included 18 patients in PPF group and 18 patients in the PRP group. Four of the 36 patients initially included in the study refused to participate, leaving 16 patients in each group. The power analysis demonstrated that 10 participants in each group would contribute to a power of 0.80 based on the change in the RM scale from preoperative to postoperative measurement in each group. The mean age of patients was 47.1 (±14.4) years and 54.7 (±14) years in the PPF and PRP group, respectively, with no significant difference (*p* = 0.156). The average days from first clinic visit until treatment were the same in both groups (23.00 vs. 23.75). The groups also had similar ratios of males to females (10:8 vs. 9:7, *p* = 0.84) in the respective groups.

In order to assess the patient’s function and pain levels, we used Visual Analog Score (VAS) for pain evaluation and Roles-Maudsley Scale (RM) for function evaluation. As shown in Table 1, both procedures demonstrated a reduction in pain 3 months after the treatment and consistently low pain scores during the follow-up year (9 to 1.62 for PPF and 8.7 to 2.4 for PRP). However, there was no significant change in VAS pain between the two groups (*p* = 0.366).

We also found a significant difference between the two groups regarding the RM score, the PPF procedure reduced the functionality scores from 3.77 to 2.08 during the sequential year. In addition, there was a significant difference in terms of change from preoperative to postoperative RM score, favoring the PRP group (−1.69 vs. −0.28, *p* < 0.0001).

On the other hand, PRP treatment showed improvement in the activity scores in that same period (3.00 to 2.72). In general, it appears to be that before treatment, the patients who proceeded to PPF had better functionality scores compared to the PRP group and lower functionality scores one-year after treatment (3.77 vs. 2.72, *p* < 0.0001) (Table 2).

Furthermore, we found that the average recovery duration which was measured as a response to physical therapy, was10.2 months vs. 37.2 in the PRP and PPF groups, respectively (*p* = 0.002).

## 8. Discussion

PRP has expanded as a form of treatment for chronic plantar fasciitis even in comparison to other more established treatments such as steroid injection and surgical intervention, which have been widely discussed in the literature for several years [16,17]. Ragab et al. and Ang et al. showed the effectiveness of intralesional steroid injection in treating CPF. However, they found it mainly has a short-term pain relief effect, lasting 4 to 8 weeks compared to PRP, which was shown to provide pain relief up to several years [18,19]. Ang et al. also reported on the complications of intralesional steroid injection such as infections, plantar fascia rupture, and heel fat pad atrophy accompanied with pain due to avascular necrosis [20]. Of note, these complications were not observed with PRP injection.

Surgical intervention (i.e., PPF) is another treatment option for plantar fasciitis that can provide pain relief and decreased postoperative morbidity at the expense of allocated time in the operating room. Samm arco et al. described postoperative complications after a surgical intervention, such as biomechanical instability, surgical wound and suture infection, weight-bearing restrictions, and prolonged absence from work [21]. In the current study, PRP showed more significant pain reduction and functional outcomes, not to mention the burden and repercussions of undergoing surgical intervention. These findings are similar to the recently published randomized control study by Shetty et al. and the randomized controlled trials meta-analysis by Ling et al. [22,23]. Although direct comparison between these two treatment methods is not widespread throughout the literature, there is cumulative evidence to support the overall effectiveness of PRP, and a strong validation of the current study’s results [18,19]. It is also consistent with our results showing that the average recovery duration, defined as a response to physical therapy, was 10.2 months in the PRP group vs. 37.3 months in the PFP group [24].

One study in the literature that directly compared PRP treatment and PPF procedure in a nature akin to us. Othman et al. compared 27 patients who received PRP injections and 23 patients who were treated by Endoscopic Plantar Fasciotomy (EPF). The authors found significant improvements in both groups regarding VAS pain and Orthopedic Foot and Ankle Society Score (AOFAS). However, they did not find a significant difference between the two groups in any of the outcome measurements as we did in our study (RM scale- 3.77 for PPF vs. 2.72 for PRP, *p* < 0.0001) [14]. In addition, the inclusion criteria differed from ours and included patients with CPF after a minimum period of 6 months with no response to conservative treatment. In the current study, the thickness of the plantar fascia was quantified using a US machine, with a threshold of a minimum of 4.5 mm thickness.

Several recent randomized controlled trials showed superiority of PRP over local injections in plantar fasciitis, in terms of pain relief and function in comparison to steroid injection [21,25,26]. Akşahin et al. reported on 60 patients who were treated with PRP injections and 40 mg Methylprednison injections for plantar fasciitis. The authors found no significant differences in VAS scores (3.4 vs. 3.93) at six months postoperatively [11]. In a prospective randomized study by Monto et al. comparing 40 patients with chronic plantar fasciitis who underwent either PRP or steroid injections. They found that PRP was more effective and durable over time than cortisone injection, using the Orthopedic Foot and Ankle Society Score (AOFAS) [12]. Another study by Jain et al. found that VAS, RM, and AOFAS scores were higher in the PRP group compared to steroid injection at 12-month follow-up postoperatively. Overall, recent studies have shown favorable results for PRP treatment over local steroid injection with improved efficacy in terms of functional score, activity score, pain reduction, and lower complication rates compared to steroid injection. These results are consistent with this study’s results, further validating its place in the literature.

## 9. Limitations

This study has several limitations, such as a relatively small cohort of patients and only one year of follow-up. However, the study benefits from a prospective analysis of prospectively collected data. In addition, though the PRP treatment was discussed and compared to local steroid injection according to the existing literature, the study lacks a third control group treated with steroid injection. Furthermore, the study was underpowered to observe differing complication rates between groups or to track changes in the plantar fascia using a US or MRI. However, our aim remained to compare patient-reported outcome scores in manners of pain and activity. Moreover, we did not guarantee that each PRP treatment was of equal concentrations of cytokines and growth factors collected in each sample, which could have introduced a bias but we are confident in our laboratory’s PRP technique to maximize the homogeneity of the samples. Finally, the patient populations maintained similar demographics and time from first visit until treatment, even though the cohorts were randomly selected, preserving the integrity of the patient selection process.

## 10. Conclusions

This study demonstrates that Platelet-Rich Plasma (PRP) treatment is a relatively safe and efficient treatment for chronic fasciitis compared to surgical intervention. In this current study, PRP demonstrated comparable pain reduction and functional outcomes with less potential complications, including response to physical therapy, that may arise with surgical intervention. Different types of PRP should further be examined in order to understand the effectiveness in treating CPF. In conclusion, we believe that PRP should be considered a viable option prior to surgical intervention.

## Figures and Tables

**Figure 1 jcm-11-06986-f001:**
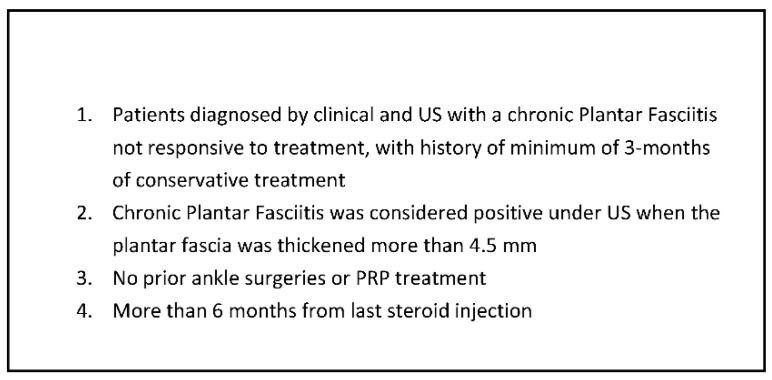
Inclusion criteria. PRP: Platelet-Rich Plasma.

**Figure 2 jcm-11-06986-f002:**
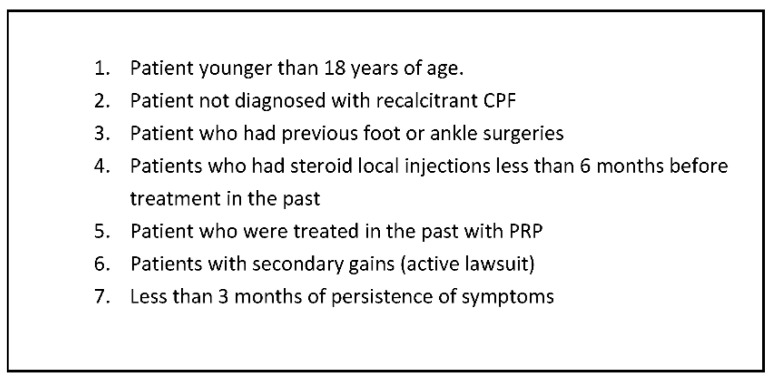
Exclusion criteria. CPF: chronic plantar fasciitis.

**Figure 3 jcm-11-06986-f003:**
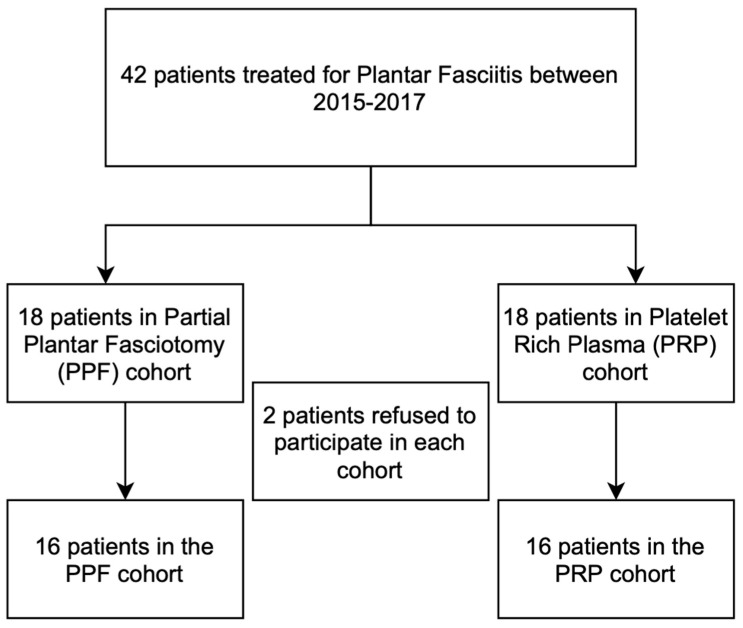
Patient selection process.

**Table 1 jcm-11-06986-t001:** Visual Analog Score (VAS) between PRP to PPF treatments.

Pain Score	Pretreatment	3-Month Post-Treatment	6-Month Post-Treatment	12-Month Post-Treatment
**PFP**	9 ± 1.7	2.38 ± 3.5	1.77 ± 1	1.62 ± 1
**PRP**	8.7 ± 1.6	3.47 ± 2.3	2.8 ± 2.7	2.4 ± 2.8
*p*-value	0.399	0.179	0.541	0.366

*p*-value refers to the comparison between PRP to PFP in each time category. VAS—Visual Analog Scale. PPF—Partial Plantar Fasciotomy. PRP—Platelets-Rich Plasma.

**Table 2 jcm-11-06986-t002:** Roles-Maudsley Scale (RM) between PRP to PPF treatments.

RM	Before Treatment	3-Moth Post-Treatment	6-Month Post-Treatment	12-Month Post-Treatment	Change from Preop to Postop
PPF	2.08	3.38	3.69	3.77	−1.69
PRP	3.00	2.64	2.64	2.72	−0.28
*p*-value	<0.0001	0.002	<0.0001	<0.0001	<0.0001

*p*-value refers to the comparison between PRP to PPF in each time category. RM—Roles-Maudsley Scale. PPF—Partial Plantar Fasciotomy. PRP—Platelets-Rich Plasma.

## Data Availability

Data can be given at the request of the authors as it is in our institution’s database.

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
