# Peer review of "Comparison of Platelet-Rich Plasma Treatment and Partial Plantar Fasciotomy Surgery in Patients with Chronic Plantar Fasciitis: A Randomized, Prospective Study"

_jcm, 2022, doi:10.3390/jcm11236986_

Round 1

Reviewer 1 Report

·         Abstract: lines 18-19. Please remove excessive spaces. Line 27, please correct: "treatment, Moreover"

·         Introduction: line 27. Please define for the first time CHPN. Lines 52-53: Please change "Platelets Rich Plasma (PRP)" to "Platelet-Rich Plasma (PRP)" (also in line 268).

·         Methods, 2.1. PRP Injection Technique: line 100: Please provide the speed in "g" for comparison with other PRP systems.

·         Methods, 2.1. PRP Injection Technique. Please, rewrite this section for better understanding. It is mandatory to offer detailed and accurate methods.

·         Methods: Please, as in other RCT the authors must follow the CONSORT STATEMENT.

·         Are characterized the PRP? Is L-PRP o P-PRP.

·         Please fill the Author Contributions and the rest of statements.

·         References: Please adapt to the in-house style of JCM.

Author Response

Comparison of Platelet-Rich Plasma treatment and Partial Plantar Fasciotomy surgery in patients with Chronic Plantar Fasciitis: A Randomized, Prospective Study

Journal of clinical medicine

We thank the reviewers for their educational comments.

According to the reviewers' comments, the entire manuscript was extensivly revised, and extensive parts were rewritten and further explained, including the Abstract

Numbered Reviewer Remark

and Manuscript Line Number

Author Response

Revised Manuscript Line

Number and Text Change

Reviewer 1:

Reviewer 1: Abstract                                                                       

1) lines 18-19. Please remove excessive spaces. Line 27, please correct: "treatment, Moreover"

Excessive spaces were removed.

The redundant spaces were removed, and the words were corrected in the the mentioned lines.

Reviewer 1: Introduction

2) line 27. Please define for the first time CHPN. Lines 52-53: Please change "Platelets Rich Plasma (PRP)" to "Platelet-Rich Plasma (PRP)" (also in line 268).

We have revised these lines.

Line 29: Chronic Plantar Heel Pain (CHPN)

Line 47: Platelet-Rich Plasma (PRP) injection

Line 234: Platelet-Rich Plasma (PRP)

Reviewer 1: Methods

3) PRP Injection Technique: line 100: Please provide the speed in "g" for comparison with other PRP systems.

We added the appropriate speed.

Line 105: Next, the tube was centrifuged in 250 g, 15 RPM/RCF (*100) for 10 minutes, using the Rotofix 32A Hettich centrifuge.

4) PRP Injection Technique. Please, rewrite this section for better understanding. It is mandatory to offer detailed and accurate methods.

The authors agree with the reviewers comments and performed the acquired adjustments.

Lines 102-112.

5) Methods: Please, as in other RCT the authors must follow the CONSORT STATEMENT.

The CONSORT Statement includes a 25-item checklist that we implemented in the study. However, we did neglect to elaborate on the randomization process. Hence we added a detailed description to the paper.

Please see the link to the CONSORT Statement site we used below:

https://www.ncbi.nlm.nih.gov/pmc/articles/PMC6398298/#:~:text=The%20CONSORT%20statement%20is%20made,and%20interpretation%20of%20the%20results

Lines 68-72

All the patients were briefed about the study's randomization process and consented to the different procedures. Only after the patient was admitted to the hospital and before taking him to the operating room did the surgeon open a sealed envelope given by the research coordinator allocating the patient to one of the two procedures. 

6) Are characterized the PRP? Is L-PRP o P-PRP

We thank the reviewer for bringing this to our attention.

Line 55:

also known as pure PRP (P-PRP) injection.

7) Please fill the Author Contributions and the rest of statements.

W thank the reviewer for this comment; the authors' names and contributions were added

Authors' contributions
Ran Atzmon, Dynai Eilig, Jeremy Dubin and Michael Drexler participated in the study design, collected the study information, and drafted the manuscript. Ran Atzmon, Matias Vidra and Alec Tavidi also collected the study information and helped perform the surgeries and revise the study. Omer Marom participated in the study design and performed all the statistical analyses. Esequiel Palmanovich was the leading surgeon and the principal investigator. All authors read and approved the final manuscript

8) References: Please adapt to the in-house style of JCM.

Reviewer 2 Report

- in intro, may want to add reference to AOFAS position statement to avoid operation on chronic heel pain until at least 6 months of failed conservative treatment

- there are many more papers regarding CPF that are not referenced in this manuscript.  obviously, don't need to reference them all but when adding a RCT to the literature, should describe other studies out there as well

- methods:  aofas statement is to avoid surgery for heel pain for at least 6 months.  patients here were enrolled at 3 months potentially.

    - how were patients randomized?

- were preop ROM/gastroc contracture evaluated?

- don't describe the PRO's to be used in the methods section

results:

- authors presents "RM" results prior to stating what "RM" is

- why were these PRO's chosen?  this should be addressed in methods section

- there are studies regarding effectiveness of gastroc recession and type procedures for relief of chronic heel pain.  ROM and presence of contracture not discussed.

Author Response

(The authors gave the same response as above.)

Author Response

Revised Manuscript Line

Number and Text Change

Reviewer 2: Introduction

1) in intro, may want to add reference to AOFAS position statement to avoid operation on chronic heel pain until at least 6 months of failed conservative treatment

For further explanation, please also see the reply to comment number 11.

We thank the reviewer for this comment; unfortunately, we failed to find a reference to the AOFAS position statement in the context of heel pain.

Please see the link to the official AOFAS's site below.

https://www.aofas.org/research-policy/position-statements-clinical-guidelines - nothing on the AOFAS official site.

That said, we agree with this statement and added a clarifying sentence.

Line 82-83

Additional exclusion criteria included patients diagnosed with CPF but failed to achieve a minimum of 3 months of conservative treatment and patients…

2) there are many more papers regarding CPF that are not referenced in this manuscript.  obviously, don't need to reference them all but when adding a RCT to the literature, should describe other studies out there as well

Thank you for this valuable input. We added two relevant articles as recommended.

Three studies have been added in lines 209-210.

Reviewer 2: Methods

3) aofas statement is to avoid surgery for heel pain for at least 6 months.  patients here were enrolled at 3 months potentially.

As cited in the paper and described in the 'Introduction' section (lines 42-46) our definition of Chronic PF was based on the Consensus Statement published by the American College of Foot and Ankle Surgeons from 2018 (PMID: 29284574), which defines chronic PF >3 months of symptoms. This can be further divided into the refectory and recalcitrant periods, with the latter consisting of symptoms presiding more than six months.

In lines 82-83, we emphasized that all patients were diagnosed with CPF, and the ones who were included in the study were only the ones who attempted a minimum of 3 months of conservative treatment. Hence, patients diagnosed with CPF but did not try a proper conservative treatment were excluded.

Line 42-46:

Establishing and defining the different types of PF

Line 67-73:

Establishing inclusion and exclusion criteria.

Lines 82-83:

Newly added clarification

4) how were patients randomized?

Please see response number 5 to reviewer 1.

5) were preop ROM/gastroc contracture evaluated?

Thank you for this excellent question. The authors contemplated this issue and were conflicted on this matter. After thorough discussions we decided to leave the gastric contractor and ROM out of this study as we explain in the newly added paragraph shown on the right.

Lines 97-100:

Though the Gastrocnemius contracture and range of motion were also assessed, it was decided not to include it in the study due to previous treatments and a lack of comparative baseline information. Hence, we felt this type of information might have a detrimental effect on the results

6) don't describe the PRO's to be used in the methods section

We thank the reviewer for this comment. A detailed description and reference of the different methods were added.

Lines 74-76: Explaining the Roles-Maudsley Scale (RM)

Lines 88-90: Explaining the Visual analog scale (VAS)

Reviewer 2: Results

7) authors presents "RM" results prior to stating what "RM" is

Please see comment number 14

8) why were these PRO's chosen?  this should be addressed in methods section

After considering several scales for our trial, we felt CAS and RM will do best since there were commonly used and easy to apply as a questioner.

Line 91-92:

Both these patient reported outcome scores were used due to their well established reputation and the fact that they're commonly used.

9) there are studies regarding effectiveness of gastroc recession and type procedures for relief of chronic heel pain.  ROM and presence of contracture not discussed.

We addressed this question in comment number 13.

Round 2

Reviewer 1 Report

Al concerns have been solved.